# Customized 3D-Printed Titanium Mesh Developed to Regenerate a Complex Bone Defect in the Aesthetic Zone: A Case Report Approached with a Fully Digital Workflow

**DOI:** 10.3390/ma13173874

**Published:** 2020-09-02

**Authors:** Marco Tallarico, Chang-Joo Park, Aurea Immacolata Lumbau, Marco Annucci, Edoardo Baldoni, Alba Koshovari, Silvio Mario Meloni

**Affiliations:** 1Department of Periodontology and Implantology, University of Sassari, 07021 Sassari, Italy; alumbau@uniss.it (A.I.L.); edoardo_baldoni@fastwebnet.it (E.B.); melonisilviomario@yahoo.it (S.M.M.); 2Division of Oral and Maxillofacial Surgery, Department of Dentistry, College of Medicine, Hanyang University, Seoul 04763, Korea; fastchang@hanyang.ac.kr; 3Private Practice, 00100 Rome, Italy; annuccimarco@gmail.com; 4Department of Implantology and Prosthetic Aspects, Aldent University, 1022 Tirana, Albania; alba.koshovari@ual.edu.al

**Keywords:** dental implants, titanium mesh scaffold, guided bone regeneration, digital workflow, anterior maxilla

## Abstract

Alveolar-ridge augmentation, anterior aesthetics, and digital technologies are probably the most popular topics in the dental-implant field. The aim of this report is to present a clinical case of severe atrophy of the anterior maxilla in a younger female patient, treated with a titanium membrane customized with computer-aided design/computer-aided manufacturing (CAD/CAM), simultaneous guided implant placement, and a fully digital workflow. A young female patient with a history of maxillary trauma was treated and followed-up for 1 year after implant placement. A narrow implant was inserted in a prosthetically driven position with the aid of computer-guided surgery. In the same surgical section, a customized implantable titanium mesh was applied. The scaffold was designed according to the contralateral maxillary outline in order to recreate a favorable maxillary bone volume. Finally, highly aesthetic, CAD/CAM, metal-free restorations were delivered using novel digital technologies.

## 1. Introduction

In the presence of a questionable tooth with an uncertain prognosis, the clinician has to decide whether to treat the tooth or to extract it and place an implant. Although preliminary results from a randomized controlled trial suggest that tooth retreatment and implant replacement had similar success rates, the natural tooth was favored for the aesthetics of the soft tissues [1]. However, in the case of hopeless or missing teeth, the neighboring teeth need to be contoured and reshaped by the removal of tooth enamel to provide space for a crown to be cemented. Dental implants are the gold standard in the treatment of partial edentulous arches [2,3,4,5]. Nevertheless, with increasing dental implant treatments, more complex cases have to be approached. Implant-supported restorations require adequate hard and soft tissue volumes at the intended implant site. A major concern in the anterior areas is the inadequate mesiodistal and/or buccolingual bone dimensions, whereas the posterior areas are frequently affected by inadequate vertical bone height. Several minimally invasive procedures have been proposed for the treatment of posterior jaws, but the treatment of anterior areas is still challenging. Guided bone regeneration using resorbable membranes and xenografts has proved to be a viable option for the rehabilitation of bone defects with dental implants [6,7,8].

Although a single, implant-supported restoration concerns a small part of the total arch, good aesthetic results require a correct diagnosis and prosthetically driven implant placement [9,10]. In recent years, computer-based, template-assisted surgery, also called guided surgery, has become more popular in dentistry [11,12,13,14]. Beside the clear benefits of the guided surgery, such as lower post-operative pain and swelling, less marginal bone loss was observed in a randomized controlled trial with 5 years of follow-up [15].

There are several major contributors to the development of prosthetically driven, implant-supported restorations. Among these are a fully comprehensive diagnosis; accurate virtual implant planning, based on a prosthetic setup; the improved accuracy of printing technologies; and global integration with other digital technologies. Implant treatments in the anterior area often require an integrated approach, including orthodontic treatments [16] or guided bone regeneration at the intended implant site [17]. Preoperative evaluation includes cross-sectional imaging (cone-beam computer tomography, CBCT), digital or digitalized study models, and integrated decision-making with patients.

The aim of this research was to present a clinical case of severe atrophy of the anterior maxilla in a younger female patient, treated with a titanium mesh scaffold customized by computer-aided design/computer-aided manufacturing (CAD/CAM), simultaneous implant placement, and a fully digital workflow.

## 2. Case Report

A 19-year-old woman with a missing maxillary right central incisor and severe bone atrophy was referred to a private dental clinic for implant-supported rehabilitation of the missing tooth. The central incisor was lost 10 years before due to a car incident that also involved the loss of the buccal bone plate and a tear of the upper lip. Due to the young age of the patient at the time of injury, a resin-bonded fixed dental prosthesis was delivered without any adjunctive treatment. However, the patient was not satisfied with this temporary solution due to continuous debonding and poor aesthetics.

During the first examination, digital radiographs, intraoral digital impressions (Medit i500, Medit Corp., Seoul, Korea), and pictures were initially taken. Intraoral digital impressions were taken with level-2 filtering and a 17.0 mm depth, following the manufacturer’s guidelines. At the intraoral examination, a high smile line, severe bone defect, and severe deep bite were noted. As a consequence, the temporary restoration has always been unstable. The patient also reported aesthetics concerns due to the severe bone defect (unsupported soft tissues, Figure 1), complicated by the high smile line.

In order to make the temporary restoration as stable as possible, it was bonded to adjacent teeth, but this worsened the aesthetics and reduced the possibility of maintaining good oral hygiene, resulting in chronic soft-tissue inflammation.

At this point, a CBCT scan was immediately taken in office. The STL files derived from the intraoral impression and the DICOM files derived from the CBCT scan were then immediately imported into dedicated software (3Diagnosys, RealGUIDE 5.0, 3DIEMME Srl, Cantù, Italy). Preliminary virtual implant planning was performed based on a virtual wax-up. The planning was shown to the patient and her parents in order to explain the need for alveolar-ridge augmentation to correct the horizontal bone defect and place the implant in the correct, prosthetically driven, position. At the CBCT analysis, no vertical defect was present, but a thickness of 4 mm at the crestal level was noted (Figure 2).

The patient understood that alveolar-ridge augmentation was the only option for resolving the aesthetic problems. A new resin-bonded, fixed dental prosthesis, as well as a bridge cemented on the reshaped neighboring teeth, was excluded for the aforementioned aesthetic reasons. Taking into consideration the young age of the patient, favorable lifestyle factors (no smoking or alcohol), and lack of pathologies, simultaneous implant placement and alveolar-ridge augmentation with a CAD/CAM-customized titanium mesh was suggested. The patient was informed about the nature of the study and gave her written consent for the surgical and prosthetic procedures and the use of all the radiologic and clinical data for publication. The principles embodied in the Helsinki Declaration of 2013 were adhered to strictly. This case report is part of previously published case-series study [17].

A new resin-bonded, fixed dental prosthesis was delivered, and a professional oral hygiene treatment was performed. Two intraoral digital impressions (Medit i500, Medit Corp., Seoul, Korea) were retaken with and without the new temporary prosthesis used as a prosthetic setup. Virtual implant planning was then performed. The implant was planned in the prosthetically driven position suitable for a screw-retained restoration, independently by the residual bone. After virtual implant planning, the project was exported, and a CAD/CAM titanium mesh was virtually designed based on the features of a marketed membrane (OssBuilder, Osstem Implant Co. Ltd., Seoul, Korea). The customized titanium mesh was designed by an expert CAD designer using dedicated software (exocad DentalCAD, exocad, Darmstadt, Germany) based on the outline and shape of the contralateral anterior maxilla (Figure 3 and Figure 4, New Ancorvis Srl, Bargellino, Calderara di Reno, Italy).

The surgical and prosthetic procedures were performed by an expert clinician (MT) certified in implant-based therapy. The patient received 2 g of amoxicillin 1 h before surgery (Zimox, Pfizer, Rome, Italy) and then 1 g twice a day for 8 days. Immediately before surgery, the patient rinsed with a 0.2% solution of chlorhexidine (Curasept, Curaden Healthcare, Saronno, Varese, Italy) for 1 min and a sterile surgical drape was applied to disinfect the surgical site. Local anesthesia was induced using articaine solution (4%) with epinephrine (1:100,000; Ubistein, 3M ESPE, Milan, Italy). A mid-crestal incision into the keratinized tissue was made using a surgical blade No. 15c, and then, a full-thickness flap was elevated. Two vertical incisions were made two teeth away from the area to be augmented. The recipient site was then carefully cleaned. The implant (Osstem TSIII 3.0 mm diameter, Osstem Implant Co., Ltd., Seoul, Korea) was inserted using a surgical template without metallic sleeves, with the aid of a surgical kit designed for narrow implants (OneMS, Osstem Implant Co., Ltd., Seoul, Korea), according to a fully guided approach. After implant placement, autogenous bone was harvested from the same surgical site, using a cortical bone collector (Safe Scraper, Micross, Meta, Reggio Emilia, Italy). A mixture of anorganic bovine bone (Bio-Oss, Geistlich Biomaterials Italia, Thiene, Vicenza, Italy) and autogenous bone, in a ratio of 1:1, was placed facing the buccal bone plate, to completely fill the bone defect (Figure 5).

The CAD/CAM, customized, 3D-printed, titanium mesh (New Ancorvis Srl. Calderara di Rone, Italy) was applied and fixed with two titanium pins (Supertack, MC Bio, Lomazzo, Como, Italy) on the buccal side. The titanium mesh was delivered nonsterile and steam sterilized before surgery using the “Universal Program” (134 °C, 2.1 bar, for 30 min; Vacuklav 40 B+ Evolution, MELAG Medizintechnik GmbH & Co., Berlin, Germany). A periosteal incision between the two vertical incisions was performed to allow the flap to be closed without any tension. The flap was finally sutured with a two-line suture. Horizontal mattress sutures (4-0 Vicryl, Ethicon, Johnson & Johnson, Pomezia, Italy) were first placed 4.0 mm from the incision line; single interrupted sutures were then placed near the edges of the flaps (5-0 polytetrafluoroethylene (PTFE), Fidenza, Italy). Vertical incisions were sutured with single, interrupted sutures. After the surgical procedure, a periapical radiograph was taken and the resin-bonded, fixed dental prosthesis was cemented.

Postoperatively, 80 mg of ketoprofen (Oki, Dompé, Milan, Italy) was prescribed as needed. Dexamethasone (Desoren, Rekah Pharmaceutical Products, Holon, Israel) was administered perioperatively (4 mg). Patient was instructed to not brush the surgical wound for 2 weeks, to rinse with 0.2% chlorhexidine (Curasept SpA, Saronno, Italy), and to follow a soft-food diet for 4 weeks.

After a 4-months uncomplicated healing period (Figure 6 and Figure 7), second-stage surgery was performed to remove the titanium mesh (Figure 8).

To accelerate the healing process, autogenous platelet-rich fibrin (PRF) was prepared and applied to the reconstructed bone.

A new temporary restoration was screw-retained to the osseointegrated implant. Three months later (Figure 9), the adjacent left-central incisor was prepared for a ceramic veneer, and a definitive digital intraoral impression was taken, with a digital scan body (Figure 10 and Figure 11, Osstem Implant Co., Ltd., Seoul, Korea) screwed onto the implant to record the implant’s position.

Physical models (master model and antagonist) were created (Model creator, Exocad) and printed (ProJet MJP 2500 Plus with VisiJet M2R-TN, 3D System Inc., Rock Hill, SC, USA). A dedicated Digital Lab Analog (Osstem Implant Co., Ltd., Seoul, Korea) and even a removable individual abutment was then assembled into the master model.

Finally, a CAD/CAM, hybrid, zirconia restoration—layered with feldspathic ceramic and bonded to a titanium abutment outside the patient’s mouth (Ti Link abutment, Osstem Implant Co., Ltd., Seoul, Korea)—was screwed onto the implant at 20 Ncm, twice (e-Driver, Osstem Implant Co., Ltd., Seoul, Korea). A feldspathic ceramic veneer was cemented onto the adjacent central incisor using PANAVIA V5 cement according to the manufacturer’s instructions (Kuraray Noritake Dental, Milan, Italy) (Figure 12 and Figure 13).

The patient was followed up for 1 year after implant placement (Figure 14, Figure 15 and Figure 16). Radiographs showed successful peri-implant bone remodeling and maintenance up to 1 year after implant placement (Figure 17).

## 3. Discussion

Adequate bone volume is essential for achieving long-term favorable aesthetics and function of an implant-supported restoration. The present case report described a clinical case of severe atrophy of the anterior maxilla in a young female patient, treated with a CAD/CAM-customized titanium membrane with simultaneous implant placement. To the best of the authors’ knowledge, this is the first case treated according to a fully digital workflow, including guided implant placement, a customized, CAD/CAM, titanium mesh, and a customized CAD/CAM, metal-free restoration. The main differences from a previous, similar report was that a customized, implantable titanium mesh scaffold was used for space maintenance, instead of a cell-occlusive barrier membrane, for the alveolar-ridge augmentation [6,7,8,9,10]. This case is noteworthy for the combination of fully digital solutions, such as guided implant placement, alveolar-ridge augmentation with a customized titanium mesh, and the fabrication of the final restorations with printed physical models.

Although the concepts of guided tissue regeneration and guided bone regeneration were originally based on excluding cells not competent to form bone, the use of cell-occlusive membranes does not appear to be an absolute requirement for bone regeneration, as shown in the presented case. Numerous authors have reported alveolar-ridge augmentation without the use of a cell-occlusive barrier membrane, particularly when biologic amplifiers such as BMP-2 are used [18].

The use of a customized, titanium mesh scaffold has been proven efficient for guiding and supporting alveolar-ridge augmentation by maintaining space for new bone growth according to a pre-established shape. Moreover, the customized titanium mesh allows a reduction in surgery time without complex adaptations. Digital technologies have revolutionized the dental workflows, improving the quality and efficiency of the treatments. In the present case, the shape and outline of the customized membrane were obtained based on the contralateral maxillary volume.

A few years ago, Park and coauthors published a preliminary evaluation of three-dimensional, preformed titanium mesh for the treatment of peri-implant alveolar bone defects [19]. Good implant success rates and prosthetic survival were observed. Moreover, the histological findings were favorable, with 80% vital bone, 5% fibrous marrow tissue, and 15% remaining allograft in a sample collected 4 months after implant placement and staged, guided bone regeneration. Although this research is only a case report, a CAD/CAM-customized titanium mesh, with a similar design and features, was used. This case opened the possibility to determine in advance the amount of bone needing to be augmented, avoiding the need for membrane-type selection. Furthermore, more extensive defects could be treated.

Tallarico et al., recently published a prospective case series evaluating the same three-dimensional, preformed titanium mesh used in combination with guided implant placement [17]; promising results were reported, extending the applicability of computer-assisted, template-guided implant placement. Guided surgery has shown excellent results in terms of accuracy. That is due to the improvement in the quality of the digital technologies. Using high-quality 3D printers, surgical templates can be produced with higher accuracy than that of those produced with old-generation templates with metallic sleeves [11,20,21]. It is the authors’ opinion that combining a prosthetically driven approach with a customized CAD/CAM titanium mesh could represent a viable treatment option, allowing for more predictable results. The main limitation of the present study is that it is only a case report with a 1-year follow-up. Nevertheless, this case may encourage further randomized controlled trials with larger sample sizes and longer follow-up.

Finally, CAD/CAM technologies are already widely used in the fabrication of dental prostheses. The main strength of the present case is probably that it was treated according to a fully digital approach. Intraoral scanners (IOSs) have become essential in everyday clinical practice, with increasing accuracy [22,23]. In the present study, an IOS was successfully used in all the surgical and prosthetic phases, from the initial diagnosis to the delivery of the final restoration. Using CAD/CAM digital technologies, a printed model was fabricated with a removable abutment for the natural tooth and a newly developed Digital Lab Analog for the implant. In this manner, a dental technician can combine traditional and innovative digital technologies to achieve good aesthetic results.

## 4. Conclusions

With the limitations of a single case report, the primary “take-away” lesson from this case report is that a fully digital approach for the treatment of aesthetic, complex bone defects in the anterior maxilla may produce satisfactory results. The second “take-away” message is that a proper learning curve, as well as well-trained team, is needed due to the seemingly extensive applications of new digital technologies.

## Figures and Tables

**Figure 1 materials-13-03874-f001:**
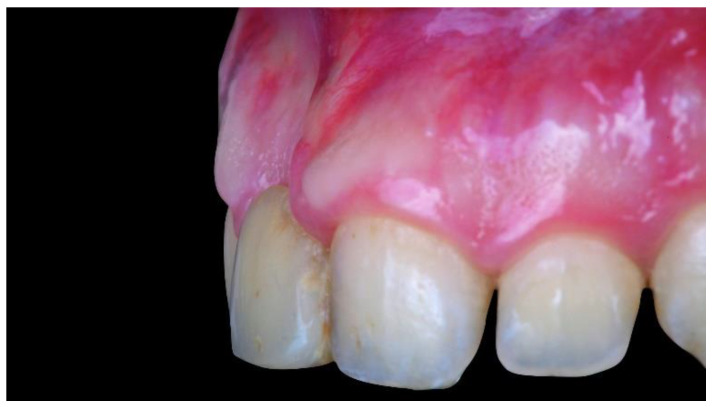
Intraoral view showing the starting position with severe bone atrophy and soft tissue deficiency.

**Figure 2 materials-13-03874-f002:**
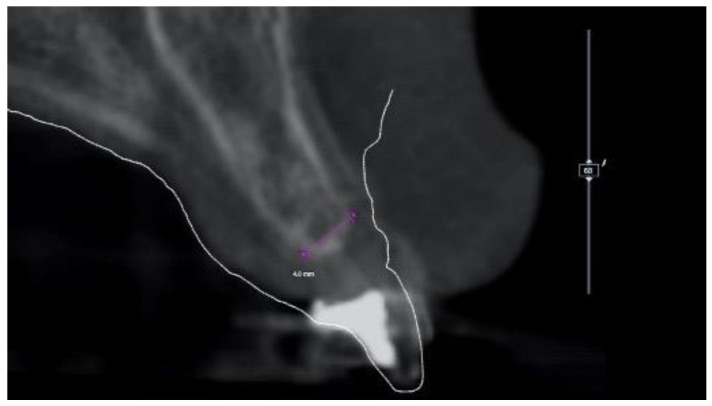
Virtual implant planning showing the implant sites with ≤4.0 mm bone in bucco-oral dimension, as measured on cone-beam computer tomography (CBCT) cross-sectional image.

**Figure 3 materials-13-03874-f003:**
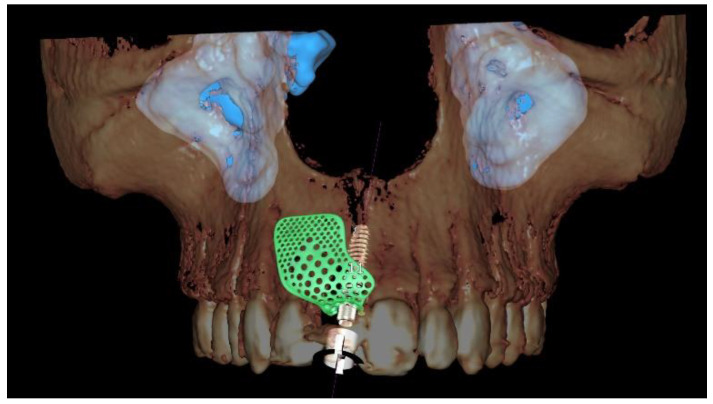
Computer-aided design (CAD) of the customized titanium mesh based on the contralateral maxillary outline and OssBuilder (Osstem Implant Co., Ltd., Seoul, Korea) features.

**Figure 4 materials-13-03874-f004:**
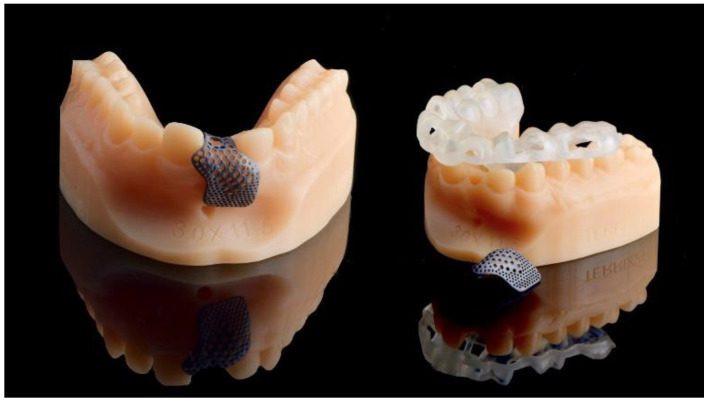
Computer-aided manufacturing (3D-printing technology) of the customized titanium mesh and the surgical template.

**Figure 5 materials-13-03874-f005:**
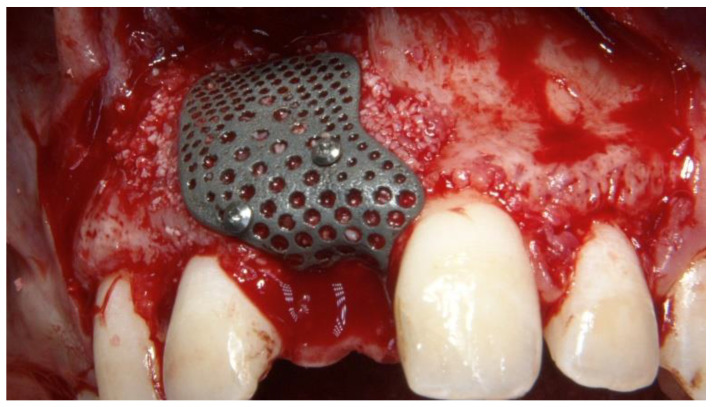
Intraoral view of the customized titanium mesh fixed using two titanium pins.

**Figure 6 materials-13-03874-f006:**
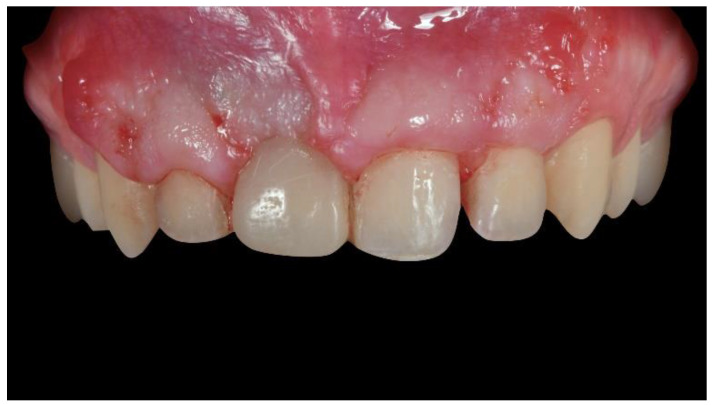
Intraoral view 3 weeks after surgery and sutural removal.

**Figure 7 materials-13-03874-f007:**
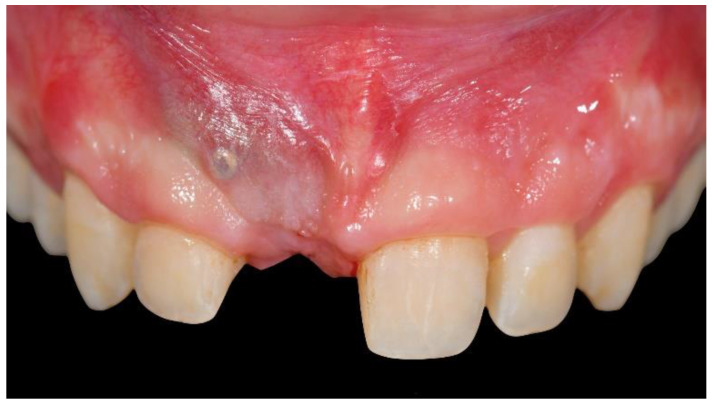
Intraoral view 4 months after uncomplicated healing.

**Figure 8 materials-13-03874-f008:**
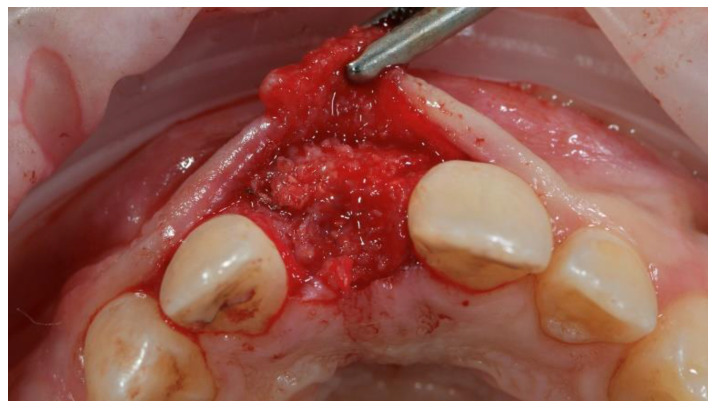
Second-stage surgery: titanium mesh removal and soft-tissue management.

**Figure 9 materials-13-03874-f009:**
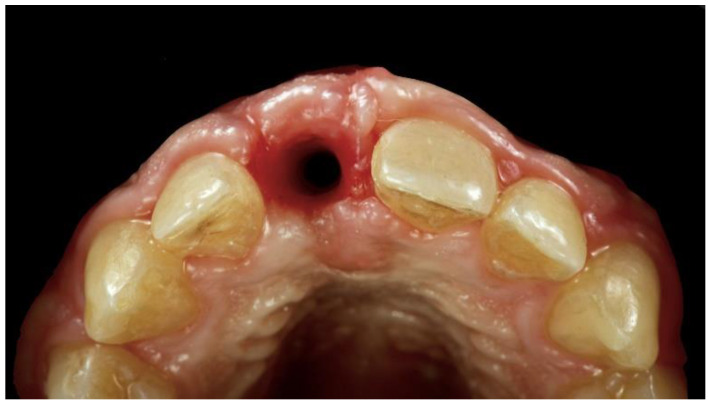
Intraoral view 3 months after second-stage surgery.

**Figure 10 materials-13-03874-f010:**
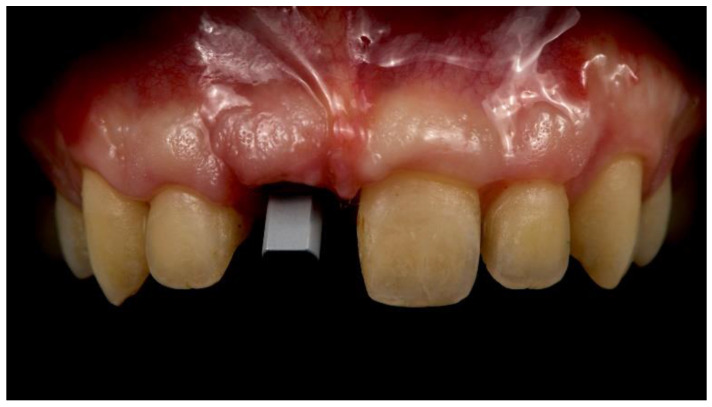
Intraoral digital impression taken using a scan body (Osstem Implant Co., Ltd., Seoul, Korea).

**Figure 11 materials-13-03874-f011:**
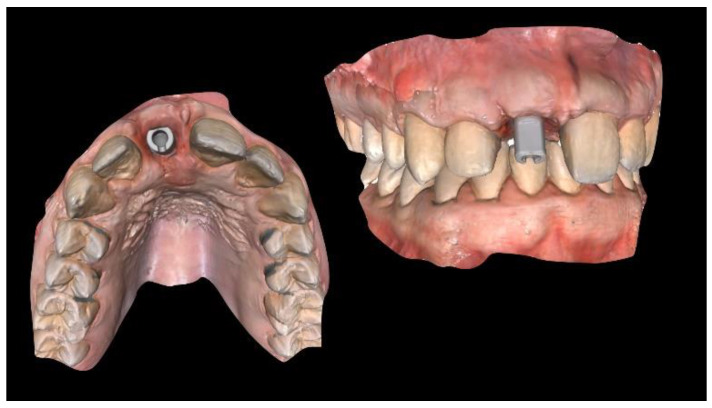
Intraoral digital impressions.

**Figure 12 materials-13-03874-f012:**
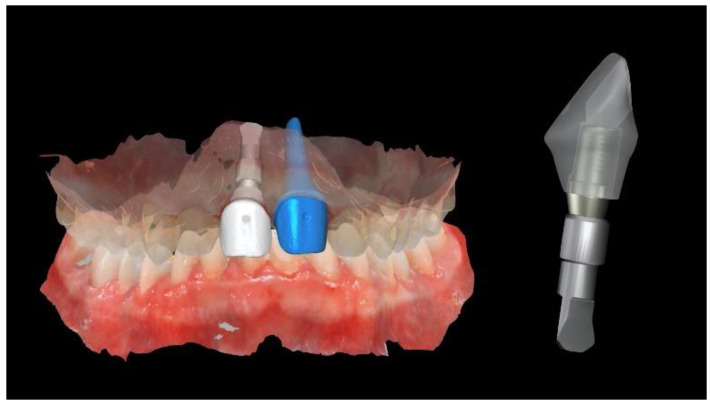
CAD of the definitive restorations. A Digital Lab Analog (Osstem Implant Co., Ltd., Seoul, Korea) was used at the implant position, while a removable abutment was used at the natural tooth position.

**Figure 13 materials-13-03874-f013:**
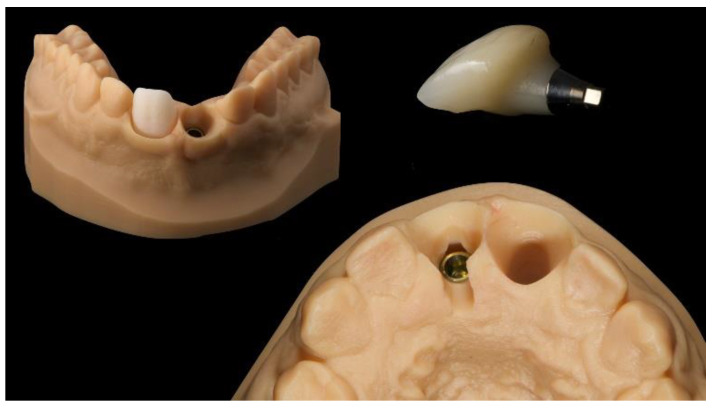
Definitive, hybrid, zirconia, screw-retained crown, cemented outside the patient’s mouth on a Ti-linked abutment (Osstem Implant Co., Ltd., Seoul, Korea). The zirconia crown was layered with feldspatic ceramic to improve the aesthetics.

**Figure 14 materials-13-03874-f014:**
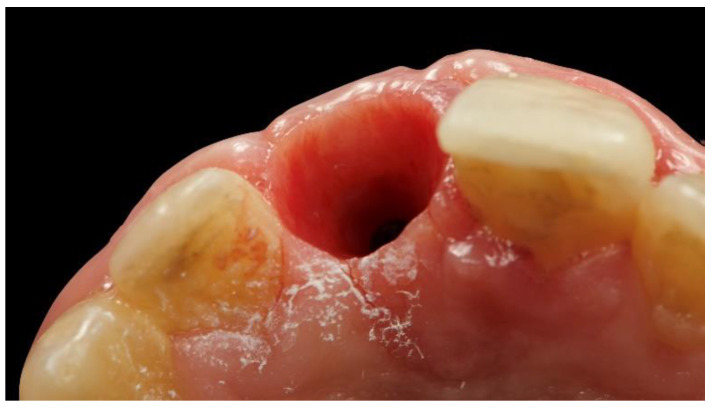
Transition zone showing good soft-tissue healing.

**Figure 15 materials-13-03874-f015:**
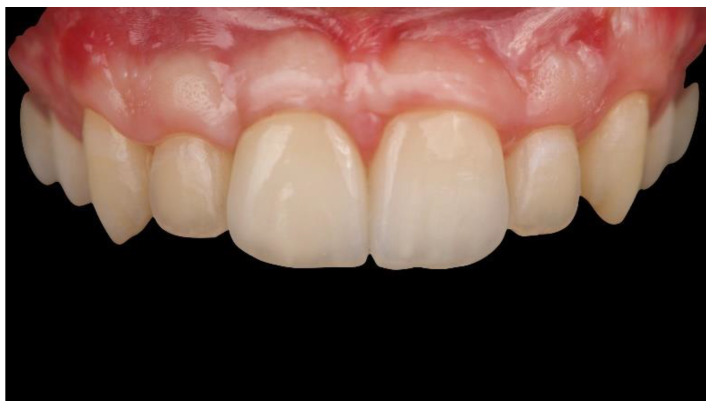
Intraoral front 1 year after implant placement (2 months after definitive-restoration delivery).

**Figure 16 materials-13-03874-f016:**
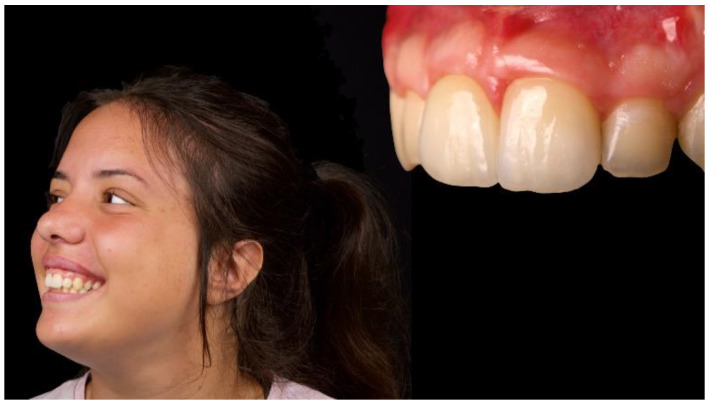
Intra- and extraoral lateral views 1 year after implant placement (2 months after definitive-restoration delivery).

**Figure 17 materials-13-03874-f017:**
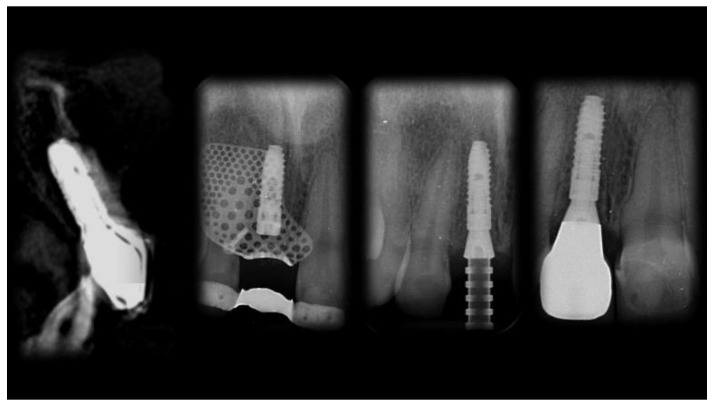
Sequence of radiographic examinations from implant placement to 1 year later.

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
