# Peer review of "Customized 3D-Printed Titanium Mesh Developed to Regenerate a Complex Bone Defect in the Aesthetic Zone: A Case Report Approached with a Fully Digital Workflow"

_materials, 2020, doi:10.3390/ma13173874_

Round 1

Reviewer 1 Report

This paper very nicely presents a clinical case of severe atrophy of the anterior maxilla in a young female treated with computer-aided design/computer aided manufacturing customized titanium membrane and guided implants placement. The case had very successful outcomes for the patient and it is important to present such a case with the detail that is presented here in order to advance this area of materials science in the dental specialty. 

Other than a few minor English edits that should be addressed with the help of an editor, this reviewer found the presentation very informative.  The details provided were very important and the images were excellent.The limitation of the work being a single case report is acknowledged and the need for further research in this area is important to note as was done so here in the manuscript.Since this case was accomplished with a completely digitally customized approach it is innovative and the "take home" message.

Other than editing the English a bit, this reviewer has no other concerns.

Author Response

Dear reviewer thank you for your revision. I completely agree and the manuscript was sent to MDPI English editing service to improve its quality.

Best regards

Reviewer 2 Report

The submitted manuscript is a well-document case report involving a printed titanium mesh device for alveolar ridge augmentation at a maxillary central incisor site.  High quality clinical photos and CBCT images are strengths of the article.  The authors are skilled clinicians and knowledgeable on the topic.  Although the methods used are not novel, a well-documented case report using advanced techniques has merit and is of interest within multiple dental specialties.  Presence of grammatical errors throughout the manuscript and use of nonstandard terminology represent major weaknesses.  Rationale for not using a cell occlusive barrier could also be addressed in the discussion section.  The article could not be published in a scientific journal in its present form.  I recommend resubmitting the article after major revision by an individual skilled in English scientific writing.    

TITLE:  The manuscript is appropriately identified as a case report in the title.  The term “aesthetic” modifies the noun “defect.”  The defect itself is not aesthetic.  Instead, I would recommend “a complex bone defect in the esthetic zone” or “a complex bone defect in the maxillary anterior.”

ABSTRACT:  “Guided bone reconstruction” is not the best term.  “Guided bone regeneration” is a term commonly applied when a cell-occlusive barrier membrane is used for space maintenance in alveolar ridge augmentation procedures.  “Alveolar ridge augmentation” may best describe the procedure presented in this report.  Technically, a case report is not research but a description of the treatment applied and the observed outcome.  “The aim of this report” could be used.  Technically, titanium is not a membrane but an implantable device  (same comment applies to the remainder of the document).  Line 21:  Plural problem—“patients” rather than “patient.”  Line 22:  Missing article—“A narrow platform dental implant” rather than “Narrow implant.”   Line 23:  Consider “a customized implantable titanium mesh device” rather than “a customized titanium mesh membrane” (same comment applies to the remainder of the document.  Line 25:  Consider “a favorable” rather than “the perfect.”

KEYWORDS:  Please check your keywords at https://meshb.nlm.nih.gov/search. 

INTRODUCTION:  Line 31: Missing article—“a questionable tooth” rather than “questionable tooth.”  Line 34:  Subject/verb disagreement—“esthetics was.”  Line 35:  “missing” rather than “missed.”  Not a good sentence.  Very unclear.  Line 41: “height” rather than “high.”  Similar grammatical and syntax problems are found throughout the document.  Line 44:  See above regarding terminology (applies to the remainder of the document as well).

CASE REPORT:  Line 69:  Rather than “Maryland bridge” the preferred term is “resin-bonded fixed dental prosthesis.”  Beautiful CBCT images, volume rendering, and clinical photos.  I recommend cropping Fig 2 to remove the extraneous portions of the image.  Thus the image of the alveolar bone could be enlarged.  Adequate description of technique.

DISCUSSION:  I recommend avoiding the word “probably.”  If this article is the first to report a fully digital workflow, from evaluation to restoration, what have previous authors shown?  I recommend citing articles that have recommended partial digital workflow and briefly describing how these reports differ from the presented method.

Although the concepts of guided tissue regeneration and guided bone regeneration were originally based on excluding cells not competent to form bone, cell occlusive membrane use does not appear to be an absolute requirement for bone regeneration, as shown in the presented case.  Numerous authors have reported alveolar ridge augmentation without use of a cell occlusive barrier membrane, particularly when biologic amplifiers such as BMP-2 are used.  The authors should address this issue in the discussion, citing appropriate literature.

Author Response

Dear reviewer, thank you for your appreciated and estensive review. I completely agree with you and I made all the required changes. Furthermore, the manuscript was sent to the MDPI English editing service for revision, in order to improve the quality of the manuscript.

I attached the file with the requested correction, before English editing.

Thanks again, 

best regards.

Round 2

Reviewer 2 Report

The manuscript is much improved after revision.

I would still prefer to avoid the term "membrane" when referring to the titanium mesh, particularly in the title.

"Customized titanium mesh device" would be a better term in my opinion.

Author Response

Dear reviewer, thanks for the two adjunctive comments.

1) I remove the term titanium mesh "membrane" thought all the text. According to other published study, and commercial terms, I used only "Titanium mesh". Sometimes, I add the term scaffold instead of device.

2) Second comment was "English language and style are fine/minor spell check required". Nevertheless, the manuscript was corrected and certified by the MDPI English service. Honestly, I don't know how to improve.

Thanks
